



# Progress in validation of rotor aerodynamic codes using field data

Koen Boorsma[1], Gerard Schepers[1], Helge Aagard Madsen[2], Georg Pirrung[2], Niels Sørensen[2],
Galih Bangga[3], Manfred Imiela[9], Christian Grinderslev[2], Alexander Meyer Forsting[2], Wen Zhong Shen[2],
Alessandro Croce[4], Stefano Cacciola[4], Alois Peter Schaffarczyk[5], Brandon Lobo[5], Frederic Blondel[6],
Philippe Gilbert[6], Ronan Boisard[7], Leo Höning[8], Luca Greco[10], Claudio Testa[10], Emmanuel Branlard[11],
Jason Jonkman[11], and Ganesh Vijayakumar[11]

[1]TNO, Petten, The Netherlands
[2]DTU, Roskilde, Denmark
[3]IAG, Stuttgart, Germany
[4]POLIMI, Milano, Italy
[5]Kiel University of Applied Sciences, Kiel, Germany
[6]IFPEN, Paris, France
[7]ONERA, Paris, France
[8]Fraunhofer, Bremerhaven, Germany
[9]DLR, Braunschweig, Germany
[10]CNR-INM, Rome, Italy
[11]NREL, Colorado, USA

**Correspondence:** K. Boorsma (koen.boorsma@tno.nl)

**Abstract.** Within the framework of the fourth phase of International Energy Agency IEA Wind Task 29, a large comparison exercise between measurements and aero-elastic simulations has been carried out featuring three simulation cases in axial, sheared and yawed inflow conditions. Results were obtained from more than 19 simulation tools originating from 12 institutes ranging in fidelity from Blade Element Momentum (BEM) to Computational Fluid Dynamics CFD and compared to state of the art field measurements from the 2MW DanAero turbine. More than 15 different variable types ranging from lifting line variables to blade surface pressures, loads and velocities have been compared for the different conditions, resulting in over 250 comparison plots. The result is a unique insight into the current status and accuracy of rotor aerodynamic modeling.

For axial flow conditions a good agreement was found between the various code types, where a dedicated grid sensitivity study was necessary for the CFD simulations. However, compared to wind tunnel experiments on rotors featuring controlled conditions, it remains a challenge to achieve good agreement of absolute levels between simulations and measurements in the field. For sheared inflow conditions, uncertainties due to rotational and unsteady effects on airfoil data result in the CFD predictions standing out above the codes that need input of sectional airfoil data. However, it was demonstrated that using CFD synthesized airfoil data is an effective means to bypass this shortcoming. For yawed flow conditions, it was observed that modeling of the skewed wake effect is still problematic for BEM codes where CFD and Free Vortex Wake codes inherently model the underlying physics correctly. The next step is a comparison in turbulent inflow conditions which is featured in IEA Wind Task 47.

Doing this analysis in cooperation under the auspices of IEA Wind has led to many mutual benefits for the participants. The large size of the consortium brought ample manpower for the analysis where the learning process by combining several





complementary experiences and modeling techniques gave valuable insights that could not be found when the analysis is carried out individually.

# 1 Introduction

Wind turbine design codes are essential for the industry to assess lifetime and the energy production before the investment is made to build a turbine prototype. The aerodynamic model is then one of the most challenging components of these codes because every aerodynamic process, in its basis, is described by means of the so-called Navier Stokes equations that cannot be solved in an analytical way where also a numerical solution of the Navier Stokes equations is out of reach in design due to extreme computational demands. The difficulty of accurate aerodynamic modelling is perhaps most convincingly illustrated by the fact that solving the Navier Stokes equations (as a matter of fact 'only' proving that a smooth solution exists) is one of the seven Millenium Prize Problems as formulated by the Clay Mathematics institute in 2000 (Fefferman, 2000). As such, every aerodynamic model inherently suffers from simplifications. For wind turbine aerodynamics, an additional difficulty arises from the fact that the computational effort for design calculations is more extreme than it is for most other applications (e.g. fixed wing aerospace), see (Schepers, 2012). This necessitates the use of engineering models that are very efficient, but also very simplified aerodynamic models based on the Blade Element Momentum (BEM) method. Obviously more advanced methods like Computational Fluid Dynamic (CFD) codes are applied too, but their use is, due to the computational demands, restricted to specific studies and load cases. From a practical point of view, the simplifications in engineering methods inevitably go together with a large uncertainty band, which is even larger for modern MW scale wind turbines. The larger uncertainties with increased rotor size are partly a result of unknown (high) Reynolds effects, where moreover the more flexible blades will lead to larger deflections and more pronounced non-linear aero-elastic behavior with unknown aerodynamic implications. Other uncertainties result from the thick(er) airfoils that are applied on large rotors and which are very difficult to model and measure accurately in an aerodynamic sense. Last but not least, the changed relation between the scales in atmospheric inflow to the scales of the turbine (blades) lead to larger uncertainties for increased rotor size.

In order to reduce the uncertainty band of aerodynamic models and to make them reliable enough for the design of cost-effective turbines, aerodynamic models need to be improved and validated with good measurements. Conventional wind turbine measurements of e.g. power and blade root bending moment lack sufficient detail for that purpose. More detailed sectional load information is necessary for a better validation and understanding. Historically, progress on this topic has taken advantage of international cooperation in research tasks under the auspices of IEA TCP Wind (IEA, 2021), leading to many mutual benefits for the participants. IEA Wind Task 14 and 18 contributed to this objective where field measurements from all over the world (some including sectional pressure measurements) were studied by an international consortium of wind energy researchers (Schepers et al., 1997, 2002). However, the main conclusion was that constant, uniform and controlled inflow conditions are necessary to make progress in this field, which led to several wind tunnel experiments with rotating rotors. Amongst these was Phase VI of the Unsteady Aerodynamics Experiment (UAE), testing a 2-bladed 10 m diameter wind turbine in the wind tunnel of NASA Ames Research Center, featuring a test section of 80 ft by 120 ft (Hand et al., 2001). The measurements taken in this





experiment have been subject of investigation in IEA Wind Task 20 (Schreck, 2008) and a blind comparison to simulations has been carried out (Simms et al., 2001). A follow up from these experiments was the European Union project 'MEXICO' (Model rotor EXperiments In Controlled cOnditions) in which ten institutes from six countries cooperated in doing experiments on an instrumented, three-bladed wind turbine of 4.5 m diameter placed in the open section of the Large Low-speed Facility (LLF) of DNW in the Netherlands. These experiments, which also featured extensive flow field measurements using Particle Image Velocimetry (PIV), featured campaigns in 2006 and 2014 and were subject of analysis in IEA Wind Task 29 phases 1 to 3 (Schepers et al., 2012, 2014; Boorsma et al., 2018). A more detailed summary of dedicated wind tunnel experiments has recently been published online (Boorsma, 2021). Although these wind tunnels have been used successfully to validate rotor aerodynamic models, translating these results to 'real life' flexible turbines in turbulent inflow conditions remains a challenge. Therefore the comparison rounds of IEA Task 29 Phase IV (Schepers et al., 2021) have focussed on newly released field measurements on a 2MW turbine from the DanAero experiment (Bak et al., 2010; Madsen et al., 2010b). The unique data from this experiment, including a description of the turbine, were made available to the participants of IEA Task 29 Phase IV so that it could form a basis for a thorough analysis. This paper presents the progress of this task, where many participants from different countries simulated the same experiment. The studies may serve as a benchmark for performing code-to-code comparison involving many participants. Suggestions will be given to improve the agreement between codes.

Section 2 presents the methodology of the comparison round, including a description of the measurements and the set-up of the comparison. Sections 3 and 4 give the results of the comparison for the two cases under investigation together with a discussion, which is followed by conclusions.

## 2 Methodology

Firstly, a description is given of the DanAero experiment. Then the set-up for the comparison rounds is given, including a summary of the simulation codes.

### 2.1 DanAero field measurement campaign

Detailed aerodynamic measurements on MW-scale wind turbines are scarce and open publications about them even more so. An exception lies in the DanAero experiment (Bak et al., 2010; Madsen et al., 2010b), which was carried out in atmospheric field conditions on a NM80 2.3MW turbine in a Danish project by Danish Technical University DTU and four industrial partners (LM Glassfiber, Siemens WindPower, Vestas and Dong Energy) in two periods from 2007 until 2010 and from 2010 until 2013. At the initiation of IEA Task 29 Phase IV in 2018, the DanAero partners agreed that the measurements as well as the model data for aerodynamic and aeroelastic modelling of the NM80 turbine could be shared with the partners participating in this task. The pitch controlled turbine features three LM38.2 blades resulting in a 80 m rotor diameter at 57 m hub height. The level of detail from the instrumentation lies far above the level of conventional wind turbine measurements. In addition to conventional power and loads measurement using strain gauges, surface pressures at four sections along a blade (at 13, 19, 30 and 37 m from the rotor center) were measured but also inflow velocities using pitot tubes, and a row of surface flush-mounted





microphones was installed at the outer part of the blade. Integrating the measured surface pressures around the sectional

airfoil contour resulted in the chord-normal and tangential pressure forces at these four stations. Meteorological measurements were performed using a mast located 313 m (3.9 diameters) in Southwesterly direction (237°) upwind from the turbine. The meteorological mast included cup and sonic anemometers at 7 heights up to 93 m. An overview is given in Figure 1. More details about instrumentation can be found in the dedicated report (Bak et al., 2013).

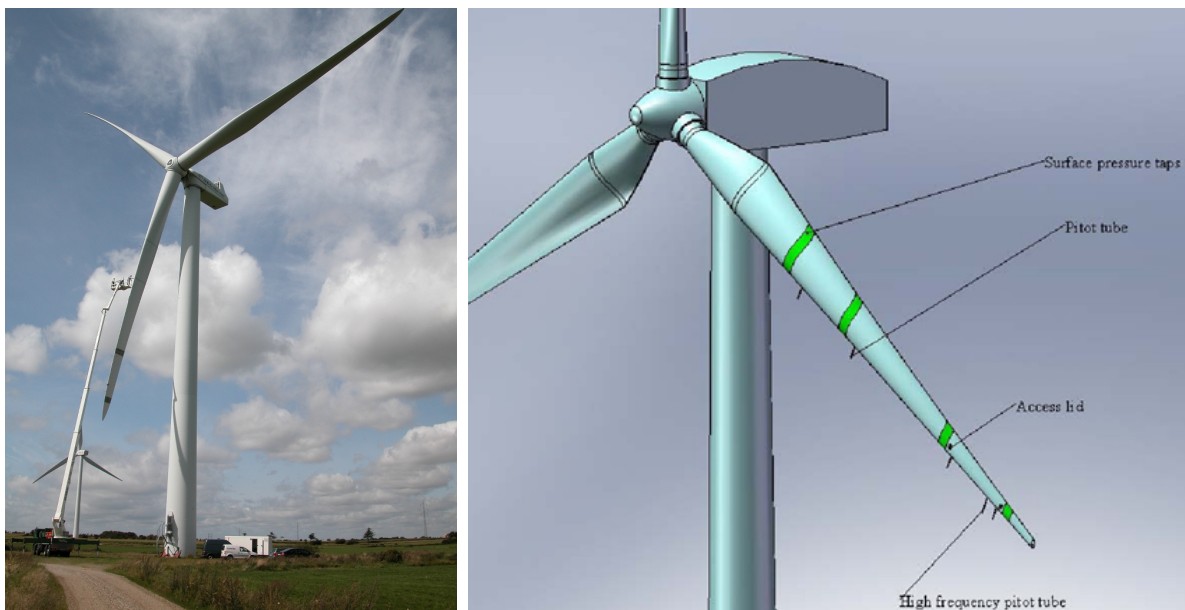

**Figure 1.** Danaero turbine and instrumentation (Bak et al., 2010)

## 2.2  Set-up

Several cases are under investigation focusing on axial and sheared-yawed inflow, which are described in more detail below. The comparison rounds mainly focus on the data obtained from the pressure measurements, i.e. pressure distributions and the derived normal and tangential force at the four instrumented stations. For the CFD modelers, the blade geometry was made available by means of a CAD file, but also a pre-processed multi-block surface and volume mesh was distributed amongst the participants. For the lifting line codes, airfoil data was prescribed. Hereto data was obtained from dedicated wind tunnel testing

of the scanned sectional airfoil geometries of the real blade, which were 3D corrected afterwards (Bak and Fuglsang, 2004).

A wide variety of over 19 different codes have been used by the participants ranging from BEM to CFD models for the rotor aerodynamics. For CFD models, both Reynolds Averaged Navier Stokes (RANS) as well as Detached Eddy Simulation (DES) formulations have been used. But also medium fidelity tools such as panel codes, actuator line (AL) and lifting line free vortex wake (LL-FVW) models are part of the comparison. An attempt to summarize the different codes has been given in Table 1.



For a more detailed description of the different simulation tools used by the participants of the comparison round, the reader is referred to the code description appendix of the final report of IEA Task 29 Phase IV (Schepers et al., 2021).

**Table 1.** High level summary of participant codes and settings

| Legend entry | Participant | Code name | Aerodynamic model | Structural model | References |
|---|---|---|---|---|---|
| Bladed4.8_BEM | DNV-GL | Bladed4.8 | BEM | multibody | (Collier, 2019) |
| DLR_TAU | DLR | Tau | RANS | rigid | (Schwamborn et al., 2006) |
| DTU_EllipSys3D | DTU | EllipSys3D | RANS | multibody (HAWC2) | (Michelsen, 1992) |
| DTU_AL_Shen | DTU | EllipSys3D | AL | modal (FLEX5) | (Sørensen and Shen, 2002) |
| DTU_AL_EllipSys | DTU | EllipSys3D | AL | rigid | (Meyer Forsting et al., 2019) |
| DTU_HAWC2 | DTU | HAWC2 | BEM | multibody | (Madsen et al., 2010a, 2020) |
| DTU_HAWC2NW | DTU | HAWC2 | BEM plus nearwake | multibody | (Pirrung et al., 2016, 2017) |
| FW_IWES_Emden | Forwind | OpenFOAM | RANS, DES | rigid | |
| IFPEN_BEM | IFPEN | DeepLines Wind$^{TM}$ | BEM (*AeroDeeP*) | multibody | (Le Cunff et al., 2013; Perdrizet et al., 2013) |
| IFPEN_VL | IFPEN | DeepLines Wind$^{TM}$ | LL-FVW (*CASTOR*) | multibody | (Bozonnet et al., 2017; Blondel et al., 2018) |
| INM_FUNAERO | CNR-INM | FUNAERO | Panel code | rigid | (Greco and Testa, 2021) |
| NREL_ED | NREL | OpenFAST | BEM (AeroDyn) | modal (ElastoDyn) | (Moriarty and Hansen, 2005) |
| NREL_VC | NREL | OpenFAST | LL-FVW (OLAF) | modal (ElastoDyn) | (Shaler et al., 2020) |
| NREL_CFD | NREL | Nalu-Wind | RANS | rigid | (Sprague et al., 2019) |
| ONERA_PUMA | ONERA | PUMA | LL-FVW | rigid | (Mudry, 1982) |
| ONERA_ElsA | ONERA | ElsA | RANS | rigid | (Cambier L., 2008) |
| PhatAero_BEM | TNO | AeroModule | BEM | multibody (Phatas) | (Boorsma et al., 2011) |
| PhatAero_AWSM | TNO | AeroModule | LL-FVW (AWSM) | multibody (Phatas) | (van Garrel, 2003) |
| PoliMi_Cp-Lambda | PoliMi | Cp-Lambda | BEM | multibody | (Bauchau et al., 2001) |
| UAS_Kiel_Tau | UAS Kiel | Tau | RANS | rigid | (Schwamborn et al., 2006) |
| USTUTT_FLOWer | USTUTT IAG | FLOWer | RANS, DES | rigid | (Kroll et al., 2000) |

    The distribution of chord-normal and tangential forces along the blades is supplied by all participants, together with rotor axial force and torque. For the CFD and panel codes, also pressure distributions are compared to the measurements. The lifting line codes that use airfoil data also supplied the distribution of so-called 'lifting line variables' (angle of attack, effective wind

speed and induced velocities), which are compared between the simulations only. To improve the comparison of the results, also non-dimensionalized values of normal and tangential force are compared. These are non-dimensionalized using undisturbed local dynamic pressure (determined from wind and local rotational speed) to allow for a solid comparison of airfoil coefficients



between experiment and simulations along the span without the complications of uncertainty in rotor induced velocities. Hence the definition for the non-dimensionalized normal force, which is equivalent for the tangential force, can be given as

$$\text{Fnc\_qc} = \frac{\text{Fn}}{0.5\rho({U_\infty}^2 + (\omega r)^2)c}, \quad \text{with} \tag{1}$$

| Fnc_qc | [-] | Non-dimensionalized chordnormal force |
| Fn | [N/m] | Chordnormal force |
| $\omega$ | [rad/s] | Rotor speed |
| $\rho$ | [kg/m$^3$] | Air density |
| $U_\infty$ | [m/s] | Wind speed |
| r | [m] | Local radius |
| c | [m] | Local chord. |

The supplied axial force and torque have been post-processed to thrust and power coefficients $\text{Cd}_\text{ax}$ and Cp using

$$\text{Cd}_\text{ax} = \frac{\text{Fax}}{0.5\rho {U_\infty}^2 \pi R^2}, \quad \text{and} \quad \text{Cp} = \frac{\text{Torque}\quad\omega}{0.5\rho {U_\infty}^3 \pi R^2}, \quad \text{with} \tag{2}$$

| Cd$_\text{ax}$ | [-] | Axial force coefficient |
| Cp | [-] | Power coefficient |
| Fax | [N] | Rotor axial force |
| Torque | [Nm] | Rotor torque |
| R | [m] | Rotor radius. |


For the axial flow case, the aerodynamic flatwise moment is deduced using a script that linearly integrates the simulated force distribution along the blade span. For the yawed and sheared case, the flatwise moment was directly supplied by the participants. The displayed experimental values have been obtained from the post-processed strain gauge measurements by
removing gravity and inertial contributions.

## 3  Case IV.1: Axial uniform inflow

The first case, IV.1, is summarized in Table 2, based on a measurement data point in summer 2009 with relatively steady and uniform inflow and constant operational conditions. It can be observed that the turbine is highly loaded with an average value of 0.38 for the axial induction factor, which is generally considered to be the turbulent wake state. For this case, the tilt angle (5°)
and tower shadow effects are neglected by the modelers, but blade pre-bend is included. The case is subdivided into case IV.1.1 where the turbine (i.e. tower, blades etc.) should be modeled rigid, and case IV.1.2 where flexibility is included. It is noted that although a comparison is made with the measurements (for which the blades are obviously flexible), the CFD simulations were mostly performed for a rigid blade.





**Table 2.** DanAero comparison cases (axial flow)

| Case nr | Model | Wind speed $U_\infty$ [m/s] | Pitch angle [°] | Rot. speed $\omega$ [rpm] | Tip speed ratio $\lambda$ [-] | Angle of attack $\alpha^\dagger$ @80%R [°] | Axial induction factor $a^\dagger$ @80%R [-] |
|---------|-------|---------|-------|-------|---------|---------|---------|
| IV.1.1 | Rigid | 6.1 | 0.15 | 12.3 | 8.4 | 4.0 | 0.38 |
| IV.1.2 | Flexible | 6.1 | 0.15 | 12.3 | 8.4 | 4.0 | 0.38 |

$^\dagger$ estimate

## 3.1 Lifting line codes

Selected *lifting line variables* comparison plots for case IV.1.2 are given in Figure 2. These variables are calculated for all codes that need the input of sectional airfoil data. The effective velocity Ueff (a composite of wind, motion and rotor induced velocities) is in good agreement between the codes, indicating that the inputted operational conditions are consistent between the codes. Small differences can be observed in the inboard region, where induction starts to play a role over the elsewhere dominant rotational velocity. The angle of attack AOA shows larger variations, caused by the differences in axial and tangential

induced velocities Ui and Vi as shown in Figures 2c and 2d. A closer look at the axial induced velocity Ui shows that, especially apparent for the participants that delivered both BEM and free vortex wake results, the free vortex wake codes (usually depicted with a dashed line) feature a roughly 10% lower induced velocity. A similar observation was made in the final report of Mexnext-III (Boorsma et al., 2018) for the New Mexico case in the turbulent wake state, which featured a high axial induction factor similar to the case under investigation here.

The resulting *loads* in terms of non-dimensionalized sectional normal and tangential force Fn and Ft are given in Figure 3. Consistent with the induced velocities and angles of attack, the agreement in Fn is fair between the computational results, with a spread around 5%. A closer look at the levels in Figure 3a shows again a difference between vortex wake codes and BEM codes, where the latter generally feature a lower loading than vortex code results. This is also reflected in the integral performance from Figures 3c and 3d (blade root flatwise moment, axial force and power). Acknowledging the earlier observed variations

in axial induced velocities, this discrepancy seems to be in contradiction with momentum theory, where a higher loading is accompanied with more induction. However this could also be attributed to uncertainties in the engineering extensions used for the turbulent wake state in which the turbine is operating for this case. Grouping the results by code type, as will be shown in section 3.3, allows to better observe the loading differences.

The comparison with measured values (obtained from the integration of the measured pressure distributions) shows the

normal force to be consistently overpredicted by approximately 10% for all radial positions. It could then be expected that



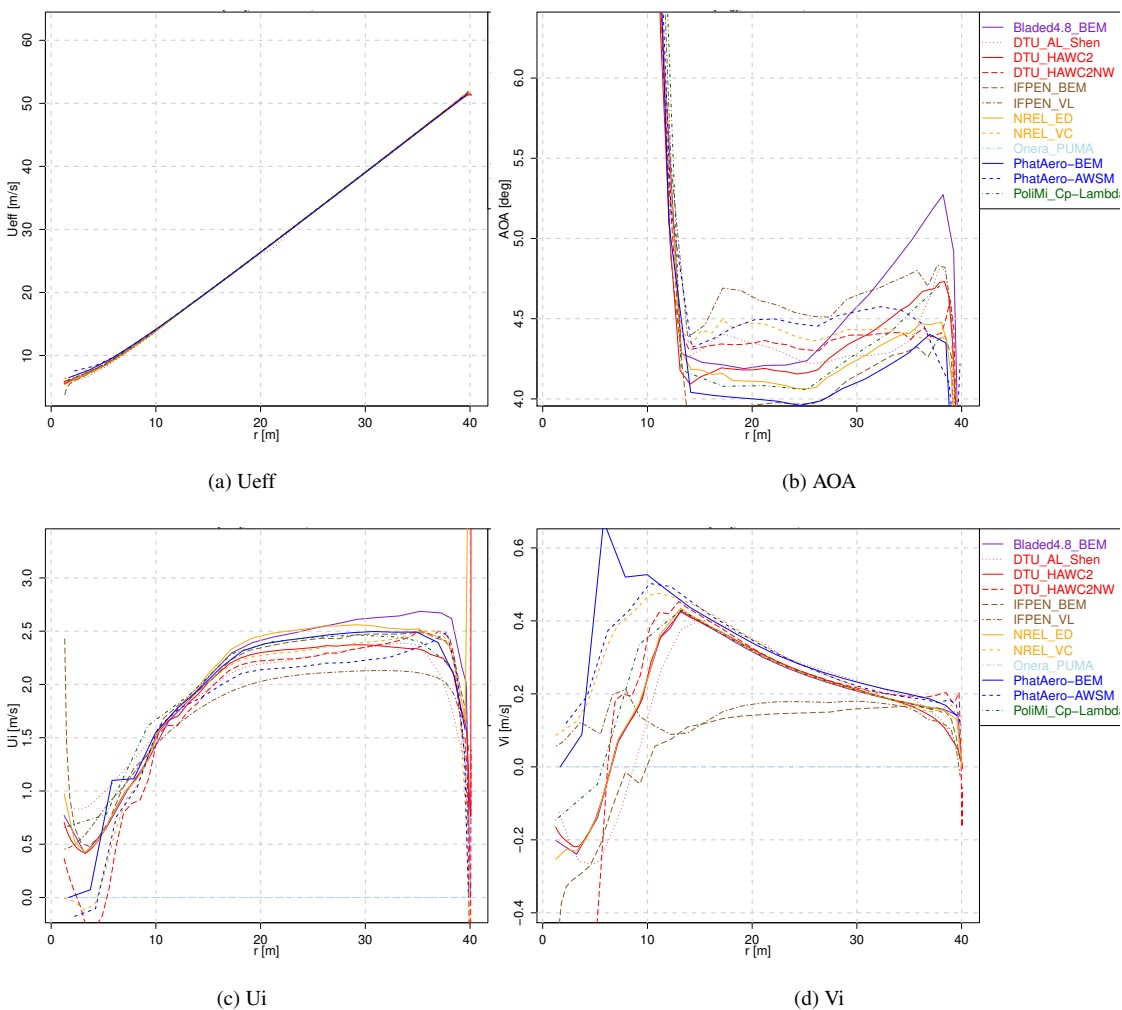

**Figure 2.** Lifting line variables for case IV.1.2

this overprediction is reflected in the measured flatwise moments from Figure 3c as well. However, the observed differences

for this moment are around $\pm 2\%$, where it should be noted that the measured flatwise moments are obtained from the strain

gauge rather than the pressure measurements. Detailed investigations of possible causes for the low experimental normal force

were carried out during IEA Task 29 Phase IV but unfortunately without success. Contrary to the normal force, the tangential

force (Figure 3b) reveals an under instead of over prediction. Here it must be noted that the measured values only contain the

contribution of pressure, whereas the simulations are based on airfoil coefficients which also contain the friction forces. For

the tangential direction, this contribution is significant (as will be shown in section 3.2) and taking this into account will result

in a better agreement between measurements and simulations. In addition to that, it is known that obtaining tangential force





from integration of a finite number of pressures is very sensitive to the distribution of the taps especially around leading and trailing edge. In summary, tangential forces obtained from pressure distributions should be interpreted with care.

Figures 3e and 3f then display the predicted ***deformations*** between the aero-elastic codes. Here it is shown that besides some outliers, most codes agree within 1% for the flapwise deformation (dx), whereas there is a bit more scatter in the edgewise deformation (dy). However, in absolute sense, we are comparing differences less than a centimeter. As some of the participants did not provide results for the flexible case (IV.1.2), several colors are visible on the zero line. Although not depicted here, the differences due to the flexibility between Case IV.1.1 and IV.1.2 are mostly noticeable in the outboard region of the blades. Because of the flapwise prebend, which is 'flattened' for the flexible case, the wind faces the blades more head-on and the radius is slightly increased resulting in slightly larger loads (and larger effective velocity, induced velocities and angle of attack) towards the tip.







(a) Fnc_qc

(b) Ftc_qc

(c) Flapwise moment Mflat

(d) Performance

(e) Flapwise deformation dx

(f) Edgewise deformation dy

**Figure 3.** Loads, performance and deformations for Case IV.1.2. Note that for the bar plots the order of the bars agrees with the order of the legend entries



## 3.2 CFD and panel codes

Starting with a comparison of the ***sectional geometries*** used by the codes, Figures 4a and 4b show the shapes of the most in-
and outboard section. Theoretically these shapes should be identical, but some offsets appear and in some cases even local
twist differences are apparent. The question remains whether these deviations are due to different ways of post-processing (e.g.
a different orientation or location of the section) or if the geometries are not identical. As the blade geometry was prescribed
by an IGES file, the latter should not be the case.

The ***pressure distribution*** comparison plots from Figure 4 show the models tend to agree well for all radial positions,
although the inviscid panel code is unable to predict separated flow in the most inboard station in Figure 4c. But at this station,
also the CFD codes struggle to match the measured pressure distribution on the suction side just before the trailing edge. For
the other three radial positions, the suction levels are 5 to 10% higher in comparison to the experiment. At r=30.20 m (Figure
4e) the linear shape of the measured pressure distribution on the pressure side just before the chordwise position of maximum
thickness differs from the rounded shape as predicted by all simulations. A separate sectional study was performed by DTU
to compare the performance between design and measured geometries, of which the first was used for this comparison round.
This however revealed very small differences in comparison to the differences shown here. Besides these small deviations,
one can conclude that generally speaking the pressure distributions are in good agreement between simulations and between
simulations and experiment.



(a) Sectional geometry at r=13.12 m

(b) Sectional geometry at r=36.78 m

(c) Pressure at r=13.12 m

(d) Pressure at r=19.06 m

(e) Pressure at r=30.20 m

(f) Pressure at r=36.78 m

**Figure 4.** Sectional geometry and pressure distributions (Case 1.1)





To further dive in the observed discrepancies between the CFD codes, a ***sensitivity study*** was carried out by DLR, DTU, UAS Kiel and IAG University of Stuttgart. Hereto their CFD results were post-processed using a common tool at IAG to improve the consistency between the CFD simulations. Additional simulations were ran using a finer mesh resolution to asses the impact of the grid resolution. The properties of the coarse and fine mesh are summarized in Table 3. Here it is noted that DTU used a fine mesh with full rotor topology, while the others used a periodic boundary condition. Although for the periodic fine mesh

**Table 3.** Coarse and fine mesh properties

| Name | Topology | Total nr of cells [M] | First cell normal size [m] | Chordwise nr of cells [-] | Spanwise nr of cells [-] | Domain size [D] | Participants |
|---|---|---|---|---|---|---|---|
| Coarse | full rotor (360°) | 14 | $1.0\times10^{-6}$ | 256 | 128 | 10 | DLR, DTU, IAG, UAS Kiel |
| Fine | full rotor (360°) | 113 | $0.5\times10^{-6}$ | 512 | 256 | 10 | DTU |
| Fine | blade periodic (120°) | 26 | $2.0\times10^{-6}$ | 256 | 200 | 10 | DLR, IAG |

the chordwise resolution is the same as the coarse mesh and the first cell normal size is even larger, the total number of cells for a full rotor equivalent of this mesh would roughly differ by a factor of 6 ($26\times3/14$). Hence the majority of the extra cells for this fine mesh are placed in the wake and the background mesh.

In addition to the grid sensitivity study, the results of the coarse and fine mesh were post-processed to observe the contribution of friction to the integrated forces. The following results can then be visualized:

cf          coarse mesh, friction and pressure forces included

cp          coarse mesh, pressure forces only


ff          fine mesh, friction and pressure forces included

fp          fine mesh, pressure forces only

The resulting comparison plots are shown in Figure 5. For the DTU result in Figures 5a and 5b, it becomes clear that the mesh refinement hardly has an influence on the results and the coarse mesh suffices. For the IAG University of Stuttgart results in Figures 5c and 5d, this conclusion does not hold and level differences up to 10% can be observed. The same holds for the DLR results (which are not shown here for the sake of briefness). While the coarse mesh is already providing grid independent results

for the incompressible DTU EllipSys3D solver, the compressible solvers require further refinement. Although the underlying pressure distributions are not shown here, these compressible codes reveal a different suction level causing the integral loads to improve for the fine mesh. Figures 5e and 5f demonstrate that refining the mesh improves the consistency between the codes significantly. Apart from the blade root region inboard of 10m span, the results are close to identical.



As expected, the effect of friction is hardly present in the normal force, which can be deduced from Figures 5a and 5c.
For the tangential force this is different, especially for the three most outboard stations, that feature moderate angles of attack
with attached flow. The friction reduces the tangential force by approximately 15%, consistent between the two code results
displayed here.



(a) Fnc, DTU

(b) Ftc, DTU

(c) Fnc, IAG

(d) Ftc, IAG

(e) Fnc, fp

(f) Ftc, fp

**Figure 5.** Effect of CFD sensitivity studies on predicted loads for Case IV.1.1





## 3.3 Model types comparison

In addition to displaying the loading results from the various codes, the supplied data also give the opportunity to calculate an
average result for each code type. Here the following code types are distinguished:

- BEM

  Blade Element Momentum methods using the prescribed airfoil data set.

- FVW

  Lifting line free vortex wake methods, also using the prescribed airfoil data.

- CFD

  Computational fluid dynamics codes, which model the rotor blade geometry and the 3D space around it. In most cases,
  the blade boundary layer is modeled as fully turbulent.

In addition to the code types listed above, there are also a panel code and actuator line codes that joined the comparison round,
but they were excluded from the averaging as the number of codes for these types are deemed too few ($<3$) to obtain valuable
statistics.

To obtain the loading averages, first the normal and tangential force are determined at the same spanwise four positions as
the instrumented sections using linear interpolation from the supplied radial distributions. A simple average $\overline{x}$ and standard
error $x_{err}$ between the supplied results of a code type are determined using

$$\overline{x} = \frac{1}{n}\sum_{i=1}^{n}(x_i) \qquad \text{and} \quad x_{err} = \sqrt{\sum_{i=1}^{n}\frac{(x_i - \overline{x})^2}{n(n-1)}}, \quad \text{with} \tag{3}$$

$x_i$      sample value

$\overline{x}$      average


$x_{err}$      standard error

$n$      number of samples.

To give an indication of the variability between the results, a partly transparent band is plotted around the average of each
code type illustrating the standard error $x_{err}$ between the supplied results of a code type. The same procedure is applied to the
flatwise moments, axial force and torque (average levels only). This process is only performed for the rigid results because the
largest number of CFD results are provided for this configuration.

The results are illustrated in Figure 6. One can conclude that the level of the normal force generally agrees within 5%
between the different code types. However the tip region shows more of a fall-off for CFD and the root region features a
sudden drop in level for the CFD results. The higher loading for free vortex wake models as discussed in section 3.1 is also
apparent from the plots. For the tangential force, the differences between the code types are larger, but in absolute sense these
are rather small. The reason for the good trend overlay between CFD and FVW opposed to BEM is not fully understood and



could as well be attributed to coincidence. The bands showing the variability between results from each code type are similar between the different types and amount to $\pm 1\%$ for the normal force and $\pm 5\%$ for the tangential force. In the previous phases of IEA Task 29, the 'human factor' would result in a larger spread between CFD results. The dedicated efforts to minimize this spread as described in section 3.2 seem to have paid off together with the fact that multiple iterations were performed to

eliminate errors.

Comparing against the measurements, the normal force is overpredicted by approximately 10% for almost all sections while the flatwise moment agrees within $\pm 2\%$ as was observed in the dedicated section discussing lifting line code and CFD results. For the discussion on the tangential force discrepancy, the reader is referred to the lifting line code discussion in section 3.1.







(a) Fnc_qc

(b) Ftc_qc

(c) Mflat

(d) Performance

**Figure 6.** Loads comparison by model types for Case IV.1.1

## 4 Case IV.2: Sheared and yawed inflow

Two cases are defined featuring a significant vertical wind shear and yaw misalignment (also including vertical shear) as summarized in Table 4, carefully selected from the available measured time series. Again it can be observed that the turbine is





highly loaded and operating in the turbulent wake state for both cases. For these cases, flexibility, tilt angle and tower shadow effects are included. It was anticipated that for many CFD contributions this was not manageable (a rigid and tilted rotor without the tower effect will suffice for CFD), but the main priority here is comparison to the measurements. Fortunately, an
investigation using a stiff and flexible aero-elastic model of the turbine in simulations indicated that the flexibility had a very small impact on the aerodynamic blade force variation. As we are studying the load variation due to shear and yaw, the results are presented as a function of rotor azimuth angle. To better compare the loads trends as a function of the azimuth angle, the mean over the rotor revolution has been subtracted from these results now showing a 'delta'. For the measurements, since results over multiple revolutions are available, the standard deviation between these measured values gives an indication of the repeatability. The standard deviation is indicated in the graphs by a grey band around the mean value.

**Table 4.** DanAero comparison cases (shear and yawed flow)

| Case nr | Shear exponent | Wind speed $U_\infty$ [m/s] | Yaw angle [°] | Pitch angle [°] | Rot. speed [rpm] | Tip speed ratio $\lambda$ [-] | Angle of attack $\alpha^\dagger$@80%R [°] | Axial induction factor $a^\dagger$@80%R [-] |
|---|---|---|---|---|---|---|---|---|
| IV.2.1 | 0.249 | 9.792 | -6.02 | -4.75 | 16.2 | 6.9 | 10.0 | 0.41 |
| IV.2.2 | 0.262 | 8.429 | -38.34 | -4.75 | 16.2 | 8.1 | 7.0 | 0.42 |

$^\dagger$ Estimate of azimuth averaged value


## 4.1 Pressure distributions

*Pressure distributions* at 0° azimuth are given for all four radial stations in Figures 7 and 8 for case IV.2.1 and IV.2.2 respectively. Also note that animations featuring four different azimuth angles (0°,90°,180° and 270°) are available from the supplement. Generally speaking the trend with azimuth is well captured, but the separated flow conditions for Case IV.2.1
(especially inboard) are a challenge for the panel code and some of the CFD codes. For both cases the measured pressure distributions at 92%R (Figures 7d and 8d) feature a dip at the suction side around 20% chord, which seem rather fierce to be introduced by transition. It is noted that for these dynamic cases involving shear most of the CFD modelers employed a different meshing strategy in comparison to the uni-axial case. More details can be found in the detailed code descriptions from the final report (Schepers et al., 2021).



**Figure 7.** Pressure distribution comparison at zero degree azimuth angle, case IV.2.1. Animations of variation with azimuth angle are available from the supplement.



**Figure 8.** Pressure distribution comparison at zero degree azimuth angle, case IV.2.2. Animations of variation with azimuth angle are available from the supplement.





## 4.2 Model types

Similar to the axial flow analysis as described in section 3.3, results between code types have been averaged to give a better overview of the differences between them. To obtain the loading averages and standard error, the same data reduction procedure is adopted as for axial flow. Since we are interested in the load variation as a function of azimuth, all supplied code results are linearly mapped onto an azimuth angle distribution with a 5° step, prior to calculating the average and standard error for each code type.

The results in terms of normal force variation are illustrated in Figure 9 and 10. Although the average level of the result is removed since we are focusing on the azimuthal variation, these were mostly in line with the differences observed in the case IV.1 in axial inflow. The trends for the sheared case in Figure 9 clearly illustrate the benefit of CFD simulations over the lifting line codes. The variability within each code type (as illustrated by the colored band around the mean results) is of the same order for all code types for this case, although there seems to be a significant variability in the amplitude of the dip around 180° azimuth as predicted by CFD codes.

The yawed case shows different trends, as it is dominated by induction rather than airfoil aerodynamics. This is illustrated by the normal force in Figure 10, with a maximum near 270° azimuth for the outer part of the blade and near 90° azimuth for the inner part. This is explained by induction effects from the skewed wake, see (Schepers, 2012): At the inner part of the blade the azimuthal load variation at yawed conditions is mainly determined by the induction from the skewed root vortex, whereas the load variation at the outer part of the blade is mainly determined by the induction from the skewed tip vortex, which gives a 180 degree different phase shift to the induction (and resulting load) variation. This skewed wake effect is inherently modelled by CFD and vortex methods by which they almost completely fall within the experimental uncertainty band, especially for the outboard sections. BEM codes rely on uncertain engineering methods to model these skewed wake effects by which the variability between BEM codes is more than two times larger than vortex and CFD codes.







(a) Fn, r/R=33%

(b) Fn, r/R=48%

(c) Fn, r/R=76%

(d) Fn, r/R=92%

**Figure 9.** Normal force Fn variation by model types, Case IV.2.1



(a) Fn, r/R=33%

(b) Fn, r/R=48%

(c) Fn, r/R=76%

(d) Fn, r/R=92%

**Figure 10.** Normal force Fn variation by model types, Case IV.2.2

## 4.3 CFD synthesized airfoil data

The poor agreement of lifting line codes with measurements for the shear case together with the encouraging CFD results instigated further investigations. IAG University of Stuttgart provided an airfoil data set synthesized from 3D rotational CFD



computations employing the azimuthal averaging method as described in (Hansen et al., 1997) and (Bangga, 2018). Two
participants then re-simulated case IV.2.1 with their BEM codes. The resulting normal force trends are given in Figure 11.

(a) Fn, r/R=33%

(b) Fn, r/R=48%

(c) Fn, r/R=76%

(d) Fn, r/R=92%

(e) Mflap

(f) Lift curves at r/R=48%

**Figure 11.** Normal force Fn variation and airfoil data using CFD synthesized airfoil data, Case IV.2.1





It is clear that for Case IV.2.1 the results featuring the CFD synthesized polars (dashed lines) outperform the results with the original polars. This indicates the earlier found discrepancies to be airfoil data related. It is noted that the airfoils are operating in the stall region and apparently the stall delay due to the 3D rotational and unsteady effects is not correctly taken into account by the orginal airfoil data. Comparing the underlying polars at r = 48%R in Figure 11f, clearly shows the stall

delay as predicted by CFD. This explains, in the case of a downward pointing blade when using the original airfoil data, the loading and induction augmentation as the lift increases for a decreasing angle of attack (when coming back from stall around 12°). It is noted that although this reversed trend is most apparent for the 48%R station, the CFD synthesized airfoil data clearly improves the agreement with measurements also for the outboard stations. This observation hints at the need for a stall delay model that works along the whole blade span instead of only the inboard part. For case IV.2.2, which is dominated by induction

aerodynamics and featuring angles of attack further away from stall, it can be shown that the CFD synthesized polars hardly impact the trend variation.

## 5 Conclusions

A large comparison exercise has been performed featuring three simulation cases in axial, sheared and yawed inflow conditions. Results were obtained from more than 19 simulation tools originating from 12 institutes ranging in fidelity from BEM to CFD

and compared to state of the art field measurements from the DanAero turbine. More than 15 different variable types ranging from lifting line variables to pressures, loads and velocities have been compared for the different conditions, resulting in over 250 comparison plots. The result is a unique insight in the current status and accuracy of rotor aerodynamic modeling.

For axial flow conditions a good agreement was found between the various code types, where a dedicated grid sensitivity study was necessary for the CFD simulations. However, compared to wind tunnel experiments like New Mexico featuring

controlled conditions, it remains a challenge to achieve good agreement of absolute levels between simulations and measurements in the field. Considerable efforts were spent on investigating possible causes in the measurements for the deviations to the simulations without success. For sheared inflow conditions, uncertainties due to rotational effects on airfoil data result in the CFD results to stand out above the codes that need input of sectional airfoil data. However, it was demonstrated that using CFD synthesized airfoil data is an effective means to bypass this shortcoming. For yawed flow conditions, it was observed that

modeling of the skewed wake effect is still problematic for BEM codes, whereas CFD and FVW codes inherently model the underlying physics correctly. The next step is a comparison in turbulent inflow conditions, which is featured in IEA Wind Task 47.

Doing this analysis in a cooperation under the auspices of IEA TCP Wind has led to many mutual benefits for the participants. The large size of the consortium brought ample manpower for the analysis where the learning process by combining several

complementary experiences and modeling techniques gave valuable insights that could not be found when the analysis was carried out individually.



*Code and data availability.* The DanAero dataset is available to participants of IEA Wind Task 47 after signing a 'light' NDA. Of the many software codes used in the comparison round, some are open source and some not.

*Author contributions.* All comparison round participants contributed by handing in simulation results. H.Aa. Madsen prepared the mea-
surement data and K. Boorsma processed simulation and measurement data by generating the comparison plots and interpreting these. All participants to the comparison rounds discussed the results together in several meetings.

*Acknowledgements.* The authors would like to thank IEA TCP Wind for facilitating the IEA Task 29 Phase IV project in their framework. The contributions of the participants to IEA Task 29 Phase IV have been funded in various national programmes, which are detailed in the corresponding final report (Schepers et al., 2021). The participant of the Danish DanAero project are acknowledged for providing the field
measurement database.



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
