# Peer review of "Progress in validation of rotor aerodynamic codes using field data"

_Wind Energy Science, 2022_

## Author Comment (AC1)

We would like to thank the reviewers for taking the time to have a look at this paper and providing their valuable suggestions.

**RC1:**

The chosen validation data were at a relatively high induction larger than 1/3. For induction factors lower than 0.3 the results when using uncorrected 2-D airfoil data normally give very good results compared to measurements. It would be nice if at least one case with a lower induction was included. The not perfect match for the loads shown in Figure 3 could be due to the 3-D correction of the airfoil data or for the BEM codes and the empirical Glauert correction. In Figure 11 some synthesized airfoil data are shown that indicate a quite different stall behavior than the prescribed ones used in the airfoil data dependent codes. Since the inflow angle and chordwise pressure distributions were measured at a few spanwise sections in the Danaero experiment it could be nice to see how this fit with the prescribed airfoil data.

Thanks for this relevant comment. Indeed the induction is rather high, but unfortunately there are no measurements available from this dataset with a lower induction, otherwise it would certainly have been included. A sentence is added to clarify this.

What is exactly meant on page 13 with the sentence, "these compressible solvers reveal a different suction level causing the integral loads to improve for the fine mesh"

This sentence is rephrased to make the point more clearly.

It is nice to see how the fully blade resolved codes give quite similar results

**Yes indeed!**

In Figure 6 it would be nice to know what tip loss model was used for the BEM and perhaps also a discussion on how the decambering effect can affect the way that loads decrease when approaching the blade tip.

**Thanks for this suggestion, a sentence is added notifying this.**

The paper use the Danaero data, but suddenly in the conclusion the New Mexico data are mentioned and the challenges to reproduce those. A more elaborate discussion of this is missing.

**Sentence rephrased to better clarify this point**

The paper is well written, but the quality of the Figures should be improved, since it is not always so easy to see the details. Also are the airfoil sections in Figure 4 a and b upside down, meaning that the suction side is the lower one ? If, yes, then it is inconsistent with the pressure plots.

Figure 4a and b have been modified accommodating this suggestion.

**RC2:**

Indeed, this is a paper, so to say "long-due". It clarifies several issues regarding aerodynamic modeling of wind turbines and its level of confidence. To my opinion the paper provides valuable information in a concise way and I would strongly recommend its publication not exactly in its present form. There are a few points that to my opinion require some attention: -In terms of modeling, it is mentioned that most CFD simulations were done in fully turbulent mode which leaves some doubt regarding the results. Please clarify this point

A sentence is added to clarify this point in section 3.2 in the paragraph about the pressure distributions.

-To my opinion, the part that concerns the choice of data to be used in models that rely on look up tables (and these all except fully resolved CFD) is important. It was also the conclusion of the AVATAR project, that when non-CFD modelling is compared to CFD, the polars must be obtained from the CFD. In the present case the improvement was substantial, to a level that would suggest to specifically include it in the conclusions and why not in the abstract.

Yes indeed the following sentence in the abstract is meant to illustrate this: 'However, it was demonstrated that using CFD synthesized airfoil data is an effective means to bypass this shortcoming'

-With regard to Fig 2, I would recommend to add a figure comparing the BEM and FV as groups in the way done in Fig 6.

Yes it would be illustrative to add this plot but to minimize the already large amount of plots it was chosen to favor the participant plot for the lifting line data and limit the code type comparison plots to the loads.

-In Fig 4 I noted that apparently there is good agreement regarding the placement of the stagnation point (this is not the case in the yawed cases). To my opinion this indicates that the flows produced by FV and CFD are similar which also seen in Fig 6 (it is said that FV and CFD agree better in terms of loads).

**The authors are not fully sure what is meant here as Fig 4 contains pressure distributions which are not predicted by the LL FVW codes.**

-The comment regarding the "poorer" grid independency trend compressible codes have in comparison to the incompressible ones, was also a conclusion drawn in AVATAR and was attributed to the pre-conditioning need in the compressible codes in low Ma conditions. The slope of convergence depended on the type of pre-conditioning

**A reference is added to the relevant AVATAR publication.**

-Are all simulations in Fig 9 and 10 rigid or only the CFD ones? How do CFD and BEM results obtained with synthetized polars compare? Perhaps instead of adding the original BEM results to add the CFD ones in Fig 11.

**Section 4 was rephrased to better clarify the simulation types between rigid or flexible. CFD results have been added to Fig. 11.**

-Although without any doubt CFD reproduces reality well, perhaps a point could be made as regards the quality over the inner region of the blade (in Fig 7 and 8 at the two inner sections the prediction of stall is not as good as in the axial case).

Yes indeed the following sentence in section 4.1 is meant to illustrate this: 'Generally speaking the trend with azimuth is well captured, but the separated flow conditions for Case IV.2.1 (especially inboard) are a challenge for the panel code and some of the CFD codes'

---

## Author Response (AR2)

Many thanks for reviewing the revised article. A response to the reported issues is given below in blue.

The paper describes a comparison between 19 simulation tools and the Dan-Aero experiment. The codes are either BEM based, Free Vortex methods or full CFD. The test cases are very little shear, significant shear and a yaw case. The BEM and the free vortex codes require as input described airfoil data and the results can depend a lot on these as also shown in the paper. The test cases unfortunately all are at a relatively high induction indicating that the overall flow is close to the turbulent wake case. The BEM results are sensitive to what Glauert models are used for the empirical CT(a) relationships when a>0.3. This should have been better described for the applied BEM methods.

Text has been added to clarify the need for an empirical relation in the turbulent wake state. For further details about each individual code the reader is referred to the code description section of the final report of this IEA Task, freely available to download.

Some of the CFD codes also have a transition model, so why only show the fully turbulent cases, since including this could have made a better agreement with the experiment.

To promote consistency between CFD results, turbulent boundary layer modeling was used. As these result already over-predict loads in comparison to the experiment, using a transition model will probably further worsen the agreement due to the corresponding lift increase at these angles of attack.

It is very weird that the geometries for the CFD based models are different.

Retrieving a sectional blade slice in a 3D pre-bended rotor geometry can result in differences due to small inconsistencies like definition of radial coordinate, angular orientation of the sectional plane and direction of the chord line. What adds to that is the fact that different software packages are being used and the corresponding 'human factor'. Hence it can be considered evident that differences appear, which is representative for what happens in real life CFD applications. This is further clarified in the text.

For the shear case shown in Figure 9 it is clear that the BEM and free vortex codes underestimate the loads for an azimuthal position of around 0, indicating that the 2-D airfoil data used stall too early as also shown in the CFD synthesized airfoil data in Figure 11f. It could have been interesting to investigate and compare with some of the classical papers showing how to correct airfoil data for rotational effects.

As described in the paper a 3D correction was applied to the default airfoil data set. Also a classical stall delay model has been used (Snel method), which showed similar results. This is now added to the text. Unfortunately these models apply a correction mainly to the inboard part of the blade, whereas it is observed that largest discrepancies can be found mid-board and also outboard. Therefore the need for a stall delay model that works along the whole blade span instead of only the inboard part is stressed in the paper.

I strongly suggest and recommend that the geometrical data and the applied blade and airfoil data for the Dan Aero experiment be made public allowing others to compare their different aerodynamic tools.

It is stressed that this data is freely shared within IEA Wind Task 47 (open for joining to all parties) after signing a 'light' NDA as required by the industrial participants of the DanAero project. This has already been indicated in the 'Code and data availability section'.